# Pharmacokinetics of Bictegravir in Older Japanese People Living with HIV-1

Akira Kawashima,[a] Hieu Tran Trung,[a,b] Koji Watanabe,[a] Misao Takano,[a] Yoshimi Deguchi,[a] Mai Kinoshita,[a] Haruka Uemura,[a] Yasuaki Yanagawa,[a] Hiroyuki Gatanaga,[a,b] Yoshimi Kikuchi,[a] Shinichi Oka,[a,b] Kiyoto Tsuchiya[a]

[a]AIDS Clinical Center, National Center for Global Health and Medicine, Shinjuku-ku, Tokyo, Japan
[b]The Joint Research Center for Human Retrovirus Infection Kumamoto University Campus, Kumamoto City, Kumamoto, Japan

Akira Kawashima and Hieu Tran Trung contributed equally to this work. Author order was determined both alphabetically.

**ABSTRACT** Bictegravir (BIC) is an integrase strand transfer inhibitor widely used in the treatment of HIV-1. Although its potency and safety have been demonstrated in older patients, pharmacokinetics (PK) data remain limited in this patient population. Ten male patients aged 50 years or older with suppressed HIV RNA on other antiretroviral regimens were switched to a single-tablet regimen of BIC, emtricitabine, and tenofovir alafenamide (BIC+FTC+TAF). Four weeks later, plasma samples were collected at 9 time points for PK. Safety and efficacy were also assessed up to 48 weeks. The median age (range) of patients was 57.5 (50 to 75) years. Although 8 (80%) had lifestyle diseases requiring treatment, no participants had renal or liver failure. Nine (90%) were receiving dolutegravir-containing antiretroviral regimens at entry. The trough concentration of BIC was 2,324 (1,438 to 3,756) (geometric mean [95% confidence interval]) ng/mL, which was markedly above the 95% inhibitory concentration of the drug (162 ng/mL). All PK parameters, including area under the blood concentration-time curve and clearance, were similar to those in young HIV-negative Japanese participants in a previous study. No correlations between age and any PK parameters were observed in our study population. No participant experienced virological failure. Body weight, transaminase, renal function, lipid profiles, and bone mineral density were unchanged. Interestingly, urinary albumin was decreased after switching. PK of BIC was not affected by age, indicating that BIC+FTC+TAF may be safely used in older patients.

**IMPORTANCE** BIC is a potent integrase strand transfer inhibitor (INSTI), widely used for the treatment of HIV-1 as part of a once-daily single-tablet regimen that includes emtricitabine and tenofovir alafenamide (BIC+FTC+TAF). Although the safety and efficacy of BIC+FTC+TAF have been confirmed in older patients with HIV-1, PK data in this patient population remain limited. Dolutegravir (DTG), an antiretroviral medication with a similar structural formula to BIC, causes neuropsychiatric adverse events. PK data for DTG have shown a higher maximum concentration ($C_{max}$) among older patients than younger patients and are related to a higher frequency of adverse events. In the present study, we prospectively collected BIC PK data from 10 older HIV-1-infected patients and showed that PK of BIC are not affected by age. Our results support the safe use of this treatment regimen among older patients with HIV-1.

**KEYWORDS** bictegravir, elderly, Japanese, pharmacokinetics

Address correspondence to Koji Watanabe, kwatanab@acc.ncgm.go.jp.

The authors declare a conflict of interest. Shinichi Oka has received research grants from ViiV Healthcare and Gilead Sciences, drug for clinical research from Gilead Sciences, and honorarium for lectures from ViiV Healthcare and Chugai Pharmaceutical. These funding sources had no impact on study design, data collection, and interpretation. The salary of Yoshimi Deguchi was partially paid from the study grant detailed under "Funding".

[This article was published on 21 February 2023 with an error in the article text. The text was corrected in the current version, posted on 1 March 2023.]

Bictegravir (BIC) is a potent integrase strand transfer inhibitor (INSTI) widely used for the treatment of HIV-1 infection as a once-daily single-tablet regimen coformulated with emtricitabine and tenofovir alafenamide (Biktarvy; BIC+FTC+TAF) (1). Although the safety and efficacy of BIC+FTC+TAF for treating older patients have been confirmed in

European populations (2), similar data for Asian populations remain limited. BIC is primarily eliminated through hepatic metabolism, with similar contributions by cytochrome P450 3A (CYP3A) and UDP glucuronosyltransferase 1A1 (UGT1A1) (3, 4). Although pharmacokinetic (PK) data for BIC have been reported in clinical trials (5, 6), limited PK data are available from studies in older patients. Dolutegravir (DTG), an antiretroviral medication with a similar structural formula to BIC, causes neuropsychiatric adverse events at a frequency associated with higher trough concentrations ($C_{trough}$) (7). Furthermore, PK data for DTG have shown a higher maximum concentration ($C_{max}$) among older patients than younger patients (8). Although neuropsychiatric adverse events are uncommon among people taking BIC+FTC+TAF, based on the results of a series of phase 3 clinical trials (9, 10), PK data for BIC in older adults infected with HIV-1 are required to confirm the safety of using BIC+FTC+TAF in this population.

We therefore evaluated the PK of BIC in older Japanese people living with HIV-1 using a prospective study. Efficacy, safety, and tolerability of BIC+FTC+TAF were also assessed over 48 weeks of treatment.

## RESULTS

**Study participants.** Twelve patients were screened for the present study, of which one patient subsequently declined to participate in the 24-h PK study. Another patient was excluded from treatment with anticancer agents for stage IV lung cancer. In total, 10 patients were enrolled in the study. The baseline demographics for the participants are presented in Table 1. The age (median [range]) of participants was 57.5 (50 to 75) years. Body mass index (BMI) was 25.5 (21.8 to 29.0) kg/m². Eight participants (80%) had lifestyle diseases requiring medical treatment, including hypertension, dyslipidemia, and hyperuricemia. Among these, dyslipidemia was reported as the most common comorbidity. No participants had renal or liver failure at inclusion, although one had congestive heart failure attributable to myocardial infarction diagnosed 1 year prior to enrollment. Nine participants (90%) were receiving a DTG-containing antiretroviral regimen (DTG+FTC+TAF or DTG-abacavir-lamivudine) at enrollment. All participants had suppressed HIV-1 RNA and high CD4 counts (519.5 [357 to 1,096])/mL at inclusion.

**PK data for BIC.** Plasma samples were collected at 9 time points from each participant for PK analysis (Fig. 1). The trough concentration ($C_{trough}$) was 2,324 (95% confidence interval [CI], 1,438 to 3,756) ng/mL, which was markedly above the expected concentration as detailed in the package insert of the drug (95% effective concentration [$EC_{95}$], 162 ng/mL) (Table 2). The maximum concentration ($C_{max}$; geometric mean [95% CI]) was 6,503 (4,399 to 9,611) ng/mL at 1.335 (0.9483 to 1.880) h after dosing. The elimination terminal half-life ($t_{1/2}$) was 18.22 (14.85 to 22.35) h, and the mean area under the plasma concentration-time curve over the last 24-h dosing interval ($AUC_{0-24}$) was 90.295 (59.268 to 127.564) h·mg/mL. These data were analogous to previously reported single-dose PK data from a young Japanese HIV-negative population (11) and slightly higher than data from HIV-positive patients in other countries (12). One patient (patient 10) showed low $C_{max}$ as well as $C_{trough}$ of BIC (Fig. 2; see Data Set S1 in the supplemental material), resulting in wide variability in the aggregated data. Therefore, we assessed the relationship between age and each PK parameter within our study population (Fig. 2) and found no correlation for any of the 6 parameters. Furthermore, no relationship between body weight or renal function and PK parameters was observed (Data Set S2).

**Clinical course after switching ART regimen to BIC+TAF+FTC.** Virological failure was not documented in any participant during the study period. One participant discontinued BIC+FTC+TAF on day 170 of treatment (scheduled visit for week 24) because of acute prerenal renal failure resulting from heatstroke. Three days after discontinuing treatment, another antiretroviral therapy (ART) regimen consisting of doravirine with FTC+TAF was initiated. HIV-RNA was not detected at the time of discontinuing BIC+FTC+TAF. All remaining participants continued BIC+FTC+TAF treatment throughout the study period. No viral rebound (>200 copies/mL) was documented during the follow-up period for all participants. A questionnaire interview was not utilized in the present study; however, no

**TABLE 1** Baseline characteristics of study participants[a]

| Patient no. | Age (yrs) | Sex | BW (kg) | BMI | Latest ART | Duration of past ART (yrs) | CD4 cell count (/mL) | HIV RNA (copies/mL) | eGFR (Cockcroft-Gault) (mL/min) | Comorbidities | Medications other than ART |
|---|---|---|---|---|---|---|---|---|---|---|---|
| 1 | 58 | Male | 73.0 | 25.3 | RPV+FTC+TAF | 20.7 | 1,096 | TND | 106.6 | None | None |
| 2 | 51 | Male | 73.0 | 27.5 | DTG+FTC+TAF | 11.3 | 498 | TND | 94.0 | NASH, dyslipidemia | Rosuvastatin (2.5 mg) |
| 3 | 74 | Male | 85.0 | 25.9 | DTG+ABC+3TC | 13.4 | 844 | <20 | 96.2 | HTN | Olmesartan (10 mg), nifedipine (10 mg) |
| 4 | 52 | Male | 71.4 | 25.6 | DTG+ABC+3TC | 12.2 | 428 | <20 | 114.8 | Hyperuricemia, dyslipidemia | Benzbromarone (25 mg), rosuvastatin (2.5 mg) |
| 5 | 58 | Male | 71.5 | 26.3 | DTG+ABC+3TC | 13.6 | 506 | <20 | 72.1 | HTN, depression | Olmesartan (10 mg), azelnidipine (8 mg), sertraline (25 mg) |
| 6 | 52 | Male | 60.0 | 21.8 | DTG+FTC+TAF | 11.1 | 533 | TND | 65.5 | None | None |
| 7 | 67 | Male | 65.7 | 24.4 | DTG+FTC+TAF | 11.7 | 357 | 163 | 74.8 | Dyslipidemia | Rosuvastatin (2.5 mg) |
| 8 | 57 | Male | 78.0 | 29.0 | DTG+ABC+3TC | 19.0 | 611 | TND | 78.9 | HTN, DM, Dyslipidemia | Atorvastatin (5 mg), telmisartan (40 mg), hydrochlorothiazide (12.5 mg), empagliflozin (10 mg), linagliptin (5 mg) |
| 9 | 50 | Male | 67.8 | 24.0 | DTG+FTC+TAF | 15.4 | 359 | <20 | 75.0 | Chronic heart failure due to old myocardial infarction, dyslipidemia | Bisoprolol (2.5 mg), enalapril (5 mg), furosemide (20 mg), rosuvastatin (2.5 mg), nicorandil (15 mg), lansoprazole (15 mg), carbocisteine (500 mg), dextromethorphan (30 mg) |
| 10 | 75 | Male | 63.7 | 24.8 | DTG+FTC+TAF | 23.9 | 815 | <20 | 126.4 | HTN, dyslipidemia, prostatic hypertrophy | Valsartan (40 mg), rosuvastatin (2.5 mg), silodosin (8 mg) |

[a]BW, body weight; BMI, body mass index; ART, antiretroviral therapy; eGFR, estimated glomerular filtration rate; RPV, rilpivirine; FTC, emtricitabine; TAF, tenofovir alafenamide; DTG, dolutegravir; ABC, abacavir; TND, target not detected; NASH, nonalcoholic steatohepatitis; HTN, hypertension; DM, diabetes mellitus; 3TC, lamivudine.

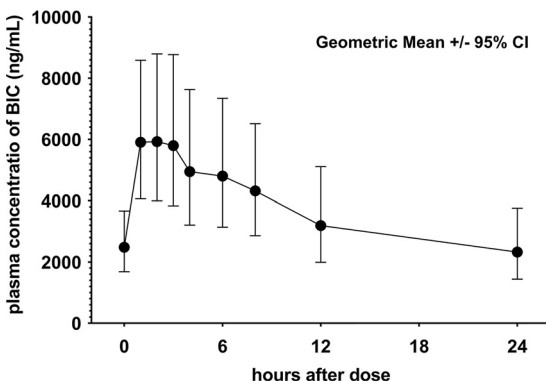

**FIG 1** Plasma concentrations of bictegravir (BIC) after oral administration of BIC+FTC+TAF. Plasma samples were collected at 9 time points, and bictegravir concentration was measured. All values are presented as geometric mean ± 95% confidence interval.

cases of neuropsychiatric symptoms or sleep disturbances were reported during the study period. Furthermore, no additional sleeping or antianxiety medications were initiated.

Adverse events and effects on organs after switching BIC+FTC+TAF were collected during the study. As described above, 1 participant (patient 9) developed prerenal renal failure at week 24 with a grade I serum creatinine level of 3.10 mg/dL. Another participant (patient 5) had an acute myocardial infarction at week 47 and underwent coronary artery bypass grafting. However, BIC+FTC+TAF was continued during the perioperative period in this participant. Regimen-related adverse events, including subjective and objective findings, were not documented in any of the participants. Body weight (BW), transaminase, renal function, lipid profiles, and bone mineral density were unchanged before and after switching to BIC+FTC+TAF (Fig. 3). Interestingly, urinary albumin was decreased after switching to BIC+FTC+TAF. Moreover, albuminuria was improved, especially among patients who had relatively high urinary albumin prior to the regimen change (patients 3, 6, and 9) (Fig. 4). Although no significant changes in urinary $\beta$2 microglobulin ($\beta$2-MG) were observed ($P = 0.1143$), urinary $\beta$2-MG was decreased in patients who had relatively high levels before switching treatment (patients 3, 6, and 10). For 6 participants, the ART regimen at inclusion contained F/TAF; however, no participants received a tenofovir disoproxil fumarate (TDF)-containing regimen. Therefore, decreased urinary albumin and $\beta$2-MG were not considered to have arisen from altered nucleotide/nucleoside reverse transcriptome inhibition but may instead have resulted from the change from DTG to BIC. However, a proof-of-concept study is warranted for further assessment of the effects on the renal tubule. In summary, BIC+TAF+FTC did not have adverse effects on any organ systems in this older patient population and potentially may have exerted a protective effect on the renal tubule, as evidenced by decreased urinary albumin and $\beta$2-MG.

## DISCUSSION

In the present study, various BIC PK parameters were assessed in older Japanese HIV-positive patients at 9 different time points (Table 2). Error ranges (95% CIs) of these variables were similar to those previously reported for BIC among younger HIV-negative

**TABLE 2** Pharmacokinetic parameters calculated by noncompartmental analysis ($n = 10$)[a]

| Data type | $C_{max}$ (ng/mL) | $C_{trough}$ (ng/mL) | $T_{max}$ (h) | $t_{1/2}$ (h) | $AUC_{0-24}$ (h·ng/mL) | $C_{L/F}$ (mL/h) | $V_F$ (L) |
|---|---|---|---|---|---|---|---|
| Geometric mean | 6,503 | 2,324 | 1.335 | 18.22 | 90,295 | 553.7 | 14.54 |
| 95% CI, lower | 4,399 | 1,438 | 0.9483 | 14.85 | 59,268 | 363.5 | 10.16 |
| 95% CI, upper | 9,611 | 3,756 | 1.880 | 22.35 | 137,564 | 843.6 | 20.83 |

[a]$C_{max}$, maximum plasma concentration; $C_{trough}$, trough plasma concentration; $T_{max}$, time of maximum plasma concentration; $t_{1/2}$, elimination half-life; AUC, area under the plasma concentration-time curve; $C_{L/F}$, capacitive loss factor; $V_F$, apparent volume of distribution; CI, confidence interval.

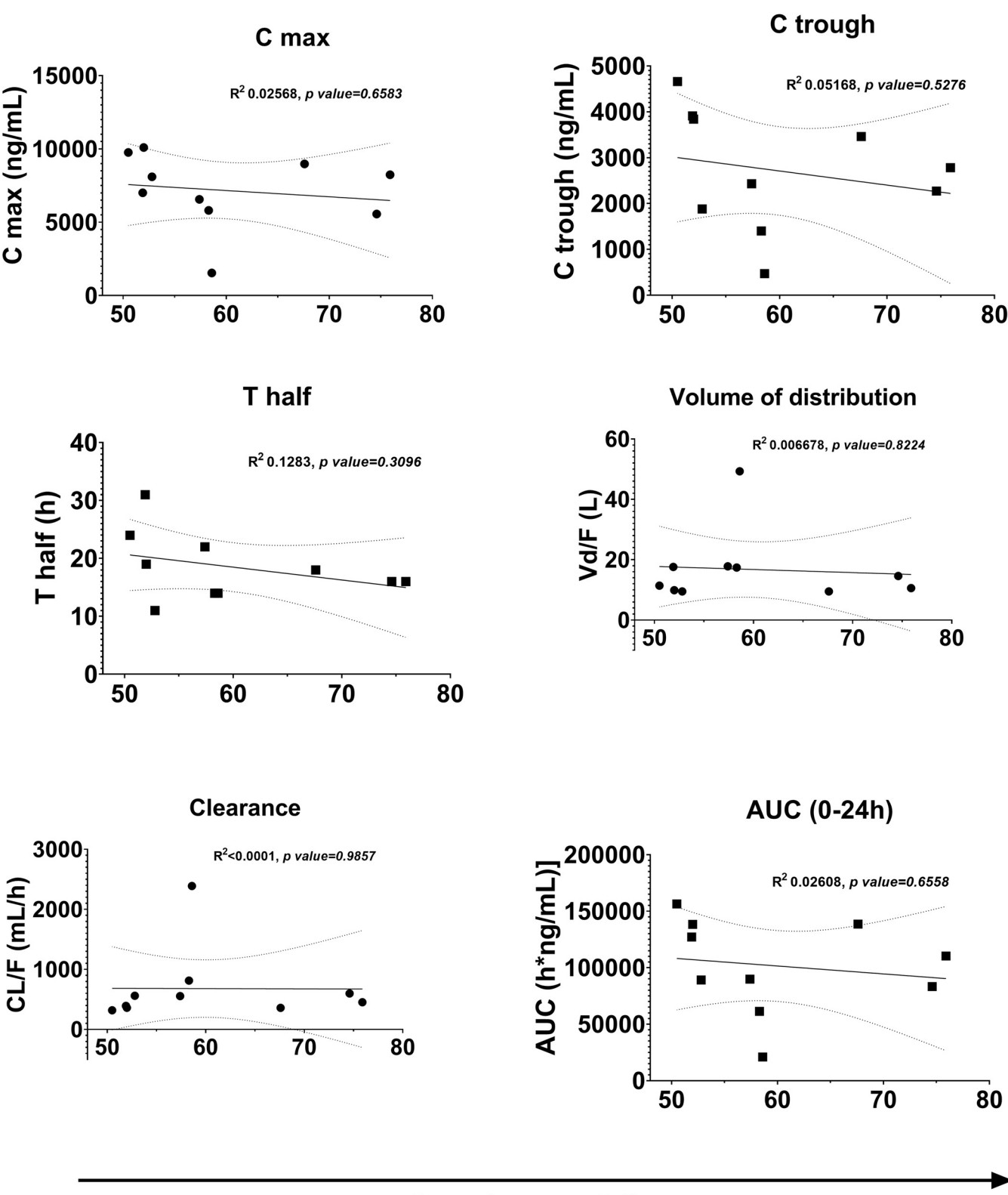

**FIG 2** Correlation between age and pharmacokinetic parameters. A simple linear regression model was used to determine the correlation between age and each PK parameter. $C_{max}$, maximum plasma concentration; $C_{trough}$, trough plasma concentration; $T_{max}$, time of maximum plasma concentration; $t_{1/2}$, elimination half-life; AUC, area under the plasma concentration-time curve; $C_{L/F}$, capacitive loss factor; $V_{d/F}$, apparent volume of distribution.

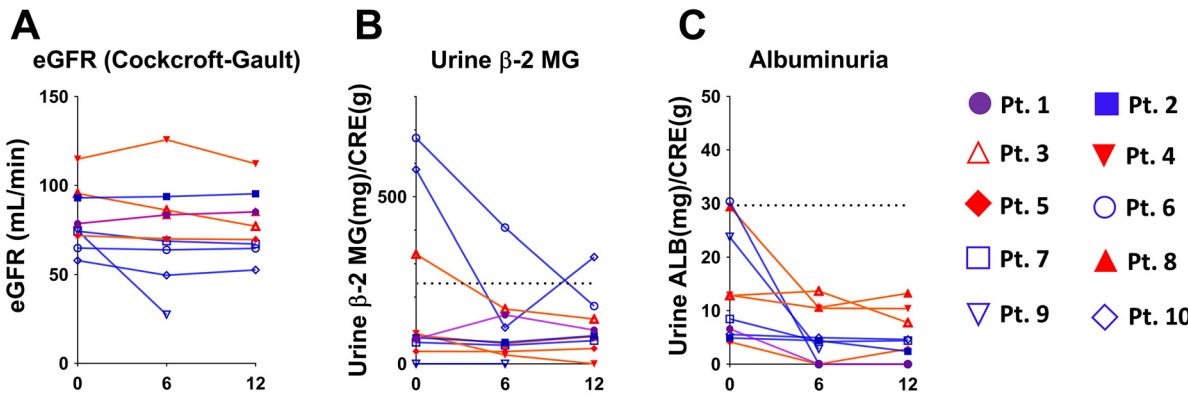

**FIG 3** Changes in clinical parameters after switching to BIC+FTC+TAF. All valuables are presented as mean ± standard error of the mean (SEM). Statistical analysis was performed by mixed-effects analysis of 3 related groups (different time points). TG, triglyceride; ALT, alanine aminotransferase.

Japanese participants (mean age, 33 years), in which $C_{max}$, $T_{max}$, $t_{1/2}$, and $AUC_{0-\infty}$ {mean (% coefficient of variation [%CV]) or median [interquartile range]} were 6,560 (18%) ng/mL, 1.00 (0.75 to 3.00) h, 17.0 (14.0 to 19.1) h, and 115,000 (21%) h·ng/mL, respectively (11). No correlation between age and any PK parameter was observed in the present study

**FIG 4** (A to C) eGFR (A), urinary β2-microgloburin (B), and urine albumin (C) at baseline, 6 months, and 12 months of treatment. Line colors represent prior regimen before enrollment. Purple, RPV+FTC+TAF; red, DTG+3TC+ABC; blue, DTG+FTC+TAF. eGFR, estimated glomerular filtration rate; MG, macrogloburin; CRE, creatinine; Pt, patient; RPV, rilpivirine; FTC, emtricitabine; TAF, tenofovir alafenamide; DTG, dolutegravir; 3TC, lamivudine.

(Fig. 2). In a previous study, theoretical PK data (physiologically based PK [PBPK] modeling of data from a phase I clinical trial) indicated that BIC PK is not significantly affected by older age (12). Our study confirmed that older age does not influence BIC concentration using prospectively collected PK data. Interestingly, AUC, $C_{max}$, and $T_{max}$ among Japanese individuals (as observed in our study and elsewhere [12]) seemed to be slightly higher and $t_{1/2}$ slightly longer than in findings from Caucasian populations, regardless of age. These results indicate that BIC PK is influenced more by ethnicity than by age. Furthermore, PK parameters may differ in the presence of comorbidities and other confounding factors. Taken together with the safety data collected during our study, our findings support the safe use of BIC as antiretroviral treatment in older patients, with no dose adjustment required for age, although a comparative study with larger sample size is required to clarify the factors affecting BIC PK.

The safety and efficacy of BIC+FTC+TAF in older patients have been reported previously (2, 13). As expected, regimen-related adverse events were not observed in the study period, although 1 patient with congestive heart failure resulting from a previous myocardial infarction developed prerenal acute renal failure attributed to heat shock at week 24 of treatment. No deterioration in liver enzymes, renal function, lipid profile, or bone mineral density was observed throughout the 48-week BIC+FTC+TAF treatment period. Interestingly, urinary albumin was decreased at 24 weeks after switching to BIC+FTC+TAF, while $\beta$2-MG was decreased in 3 patients with $\beta$2-MG higher than the normal range at baseline. BIC requires no dose adjustment in patients with creatinine clearance of 15 to 29 mL/min (14) and has been shown to be safe in patients on hemodialysis (15). However, favorable effects of BIC on the renal tubule have not been reported. A potential protective effect of BIC on the renal tubule should be assessed in a future study with a larger sample size of patients with renal dysfunction and/or proteinuria.

This study had some limitations. First, our study population contained patients with HIV-1 aged 50 years or older but did not include younger patients in the same study for comparison. We instead compared our data with those from a previous study in younger HIV-negative individuals (11) because no previous PK data from an HIV-positive younger Japanese cohort were available. Furthermore, no female Japanese patients were included in the present study because the proportion of women with HIV, especially older women, is extremely small in Japan (16). However, data from previous studies (11, 12) and the BIC package insert (4) show that PK parameters of BIC are not affected by sex. Second, only Japanese individuals were included in this study. AUC and $C_{max}$ have been shown to be slightly higher, and $t_{1/2}$ slightly longer, in Japanese participants than in Caucasian individuals in a previous phase I PK study (11). Furthermore, AUC and $T_{max}$ in the present study were marginally higher than those calculated by PBPK modeling in the SWISS cohort of older patients (12). Considered together, BIC concentrations may be elevated among Asian patients.

In conclusion, this prospective switching study of BIC+FTC+TAF confirmed that PK of BIC is not affected by age and supports the safe use of this treatment regimen among older Japanese patients with HIV-1.

## MATERIALS AND METHODS

**Study participants and ethics.** A prospective cohort study was performed at the AIDS Clinical Center, National Center for Global Health and Medicine, which is the central HIV treatment facility in Japan. Japanese patients whose HIV RNA had been suppressed with combination antiretroviral treatment without BIC were recruited according to the following inclusion criteria: males with HIV-1 infection aged 50 years or older, creatinine clearance >30 mL/min, body mass index (weight [kg]/height [m]) (2) of 18 to <35; no known resistance to INSTI, no prior virologic failure (HIV RNA > 200 copies/mL) on INSTI, and able to take medication in the morning for 24-h PK studies. We excluded patients meeting the following criteria (enrollment confirmation form; see Data Set S3 in the supplemental material): history of liver cirrhosis, inability to give informed consent, urine protein level equal to or more than 2+, and use of concurrent medication which might interact with BIC, including metformin and other dietary supplements containing divalent or trivalent cations.

This research was conducted in accordance with the Declaration of Helsinki and national and institutional standards. The study protocol was approved by the institutional review board for clinical research of the National Center for Global Health and Medicine (approval no. NCGM-G-003461-00). Written

informed consent was obtained from all participants prior to enrollment. The study protocol was registered at UMIN-CTR (UMIN00004113).

**Study procedures.** Patients were screened before enrollment by at least two physicians and two clinical research coordinators. Antiretroviral therapy was switched to standard-dose BIC+FTC+TAF at enrollment. Patients were followed for at least 48 weeks, with visits at weeks 4, 12, 24, 36, and 48. Visits and measurements are detailed below:

Screening: Written informed consent, baseline characteristics, HIV-RNA serum creatinine.

BIC+FTC+TAF were switched and HIV-RNA, CD4 cell count, serum creatinine, alanine aminotransferase, body weight, bone mineral density, urinary markers (urinary $\beta$2 microglobulin and urinary albumin) and lipid markers (triglyceride and low- and high-density lipoprotein [LDL and HDL, respectively] cholesterol) were measured at enrollment.

HIV-RNA, CD4 cell count, serum creatinine, alanine aminotransferase, body weight, urinary markers (urinary $\beta$2 microglobulin and urinary albumin) and lipid markers (triglyceride, LDL and HDL cholesterol) were measured at week 4.

HIV-RNA, CD4 cell count, serum creatinine, alanine aminotransferase, body weight, urinary markers (urinary $\beta$2 microglobulin and urinary albumin) and lipid markers (triglyceride, LDL and HDL cholesterol) were measured at week 12.

HIV-RNA, CD4 cell count, serum creatinine, alanine aminotransferase, body weight, bone mineral density, urinary markers (urinary $\beta$2 microglobulin and urinary albumin) and lipid markers (triglyceride, LDL and HDL cholesterol) were measured at week 24.

HIV-RNA, CD4 cell count, serum creatinine, alanine aminotransferase, body weight, urinary markers (urinary $\beta$2 microglobulin and urinary albumin) and lipid markers (triglyceride, LDL and HDL cholesterol) were measured at week 36.

HIV-RNA, CD4 cell count, serum creatinine, alanine aminotransferase, body weight, bone mineral density, urinary markers (urinary $\beta$2 microglobulin and urinary albumin) and lipid markers (triglyceride, LDL and HDL cholesterol) were measured at week 48.

**Sampling for bictegravir PK studies.** Study participants underwent 24-h PK sampling under hospitalization at least 4 weeks after switching to BIC+FTC+TAF. Participants fasted for 10 h prior to oral administration of BIC+FTC+TAF, and peripheral blood samples were collected into heparin-containing tubes at 0 (trough concentration), 1, 2, 3, 4, 6, 8, 12, and 24 h after administration. Samples were centrifuged at $1,500 \times g$ for 5 min, and the resulting plasma (supernatant) was stored at $-80°C$ until the measurement of drug concentration.

**Measurement of plasma concentration of BIC and PK parameters.** BIC and raltegravir-$d_3$ (internal standard [IS]) were purchased from Toronto Research Chemicals (Toronto, ON, Canada). Human blank plasma was obtained from Tennessee Blood Services (Memphis, TN, United States). Plasma samples (30 $\mu$L) were deproteinized with methanol using FastRemover (GL Sciences, Tokyo, Japan). The Nextera X2 liquid chromatography (LC) system (Shimadzu, Kyoto, Japan) and Triple Quad 5500 mass spectrometer (AB Sciex, Framingham, MA) equipped with a turbo electrospray ionization source were used to measure plasma concentrations of BIC. Chromatographic separation was achieved on an InertSustain $C_{18}$ column (100 mm by 2.1 mm inside diameter [i.d.]; particle size, 3.0 $\mu$m; GL Sciences) using 20 mmol/L ammonium formate/methanol (7:3, vol/vol) as the mobile phase at a flow rate of 0.45 mL/min. The injection volume was 2 $\mu$L, and the run time was 5 min. The mass spectrometer was operated in positive electrospray ionization mode. The mass transitions were $m/z$ 450.1→289.2 for BIC and $m/z$ 448.2→364.2 for IS. Calibration curves for BIC were linear in the range of 0.5 to 1,250 ng/mL ($r^2 = 0.99$). The intra- and interday precision and accuracy for BIC in plasma were coefficient of variation (CV) within 15.0%, respectively. The PK parameters were determined by noncompartmental analysis using Phoenix WinNonlin version 8.2 software (Certara, Princeton, NJ).

**Statistical analysis.** A simple linear regression model was used to investigate the correlation between age and each PK parameter. Body weight and other laboratory markers were compared by mixed-effects analysis of 3 related groups (baseline and 6 and 12 months after switching to BIC+FTC+TAF). All statistical analyses were performed using GraphPad Prism 9.1.0 software (GraphPad Software, La Jolla, CA, USA).

## SUPPLEMENTAL MATERIAL

Supplemental material is available online only.

**SUPPLEMENTAL FILE 1**, PDF file, 0.9 MB.

## ACKNOWLEDGMENTS

We are grateful to the staff at the AIDS Clinical Center at the National Center for Global Health and Medicine for their contribution to the present study. We thank Clare Cox from Edanz (https://jp.edanz.com/ac) for editing a draft of the manuscript.

This work was supported by Gilead Sciences, Inc. USA (study number IN-JP-380-5724).

S.O. has received research grants from ViiV Healthcare and Gilead Sciences, a drug for clinical research from Gilead Sciences, and honorarium for lectures from ViiV Healthcare and Chugai Pharmaceutical. These funding sources had no impact on study design, data

collection, and interpretation. The salary of Y.D. was partially paid from the study grant detailed under "Funding."

All authors significantly contributed to this study. Study design was conceptualized by K.W. and S.O., who were overall responsible for the present study. A.K. and K.W. wrote the manuscript. H.T.T., M.K., and K.T. performed the PK data analysis. Y.D. and M.T. performed the data collection and validation. Y.Y. and H.U. were involved in the recruitment and management of study participants. H.G., Y.K., and S.O. provided study supervision. All authors have approved the final version of the manuscript.

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
