## [Reviewer comments · Microbiology Spectrum]

Microbiology Spectrum

Pharmacokinetics of bictegrovir in older Japanese people living with HIV-1

Akira Kawashima, Hieu Trung, Koji Watanabe, Misao Takano, Yoshimi Deguchi, Mai Kinoshita, Haruka Uemura, Yasuaki Yanagawa, Hiroyuki Gatanaga, Yoshimi Kikuchi, Shinichi Oka, and Kiyoto Tsuchiya

Corresponding Author(s): Koji Watanabe, Kokuritsu Kenkyu Kaihatsu Hojin Kokuritsu Kokusai Iryo Kenkyu Center

Review Timeline:

Submission Date:	December 11, 2022
Editorial Decision:	December 28, 2022
Revision Received:	January 18, 2023
Accepted:	January 31, 2023

Editor: Takamasa Ueno

Reviewer(s): Disclosure of reviewer identity is with reference to reviewer comments included in decision letter(s). The following individuals involved in review of your submission have agreed to reveal their identity: Chien-Ching Hung (Reviewer #1); Doreen Kamori (Reviewer #2)

Transaction Report:

DOI: <https://doi.org/10.1128/spectrum.05079-22>

December 28, 2022

Dr. Koji Watanabe
Kokuritsu Kenkyu Kaihatsu Hojin Kokuritsu Kokusai Iryo Kenkyu Center
AIDS Clinical Center
1-21-1 Toyama
Shijuku-ku
Tokyo 162-8655
Japan

Re: Spectrum05079-22 (Pharmacokinetics of bictegrovir in older Japanese people living with HIV-1)

Dear Dr. Koji Watanabe:

Important issues were raised by the reviewers although the study is interesting. In particular, I agree with the reviewers that the issues of the small sample size (Reviewer 1) and the absence of female samples (Reviewer 2) are necessary to be adequately addressed in the revised version.

Link Not Available

Sincerely,

Takamasa Ueno

Journals Department
Reviewer comments:

Reviewer #1 (Comments for the Author):

The authors investigated the pharmacokinetics of bictegrovir among 10 Japanese people living with HIV-1 (PLWH) who were men aged 50 years or older 4 weeks after stable switch to coformulated bictegrovir, emtricitabine, and tenofovir alafenamide. The study is interesting; however, the sample size is too small. I have a few comments for the authors to consider.

Major comments:

1. While the study was designed as a pharmacokinetic investigation, the sample size of 10 patients is too small. As shown in Figure 2, the variability of trough concentration of bicitegravir is wide among the enrolled participants aged between 50 years and 80 years.
2. Did the author investigate the intracellular concentration of tenofovir diphosphate among the participants?
3. The conclusion on safety and efficacy among PLWH aged 50 years or older could not be safely reached from this single-arm study consisting of only 10 PLWH. Similarly, I am afraid that the conclusion that PK parameters were not correlated with the age of the participants enrolled can only be made with a larger sample size.
4. In the cited dolutegravir study (ref 8) regarding the PK parameters and adverse effects, PLWH who were aged 60 years or older were enrolled; questionnaire interview was conducted to assess the neuropsychiatric aspects and sleep quality. In this current study, no such assessment was conducted.
5. The authors are encouraged to provide more information on dietary supplements of the participants, not just those prescribed medicines for chronic diseases. Many dietary supplements taken by the participants might contain divalent or trivalent cation and might not be known to the treating physicians unless specifically inquired.
6. The tubular functions in this study were assessed by albumin:creatinine ratio. The author are encouraged to provide data on urinary beta-2-microglobulin:creatinine ratio, which will be a better parameter for assessment of tubular function than urinary albumin:creatinine ratio.
7. There were six participants who were receiving TAF-containing regimens before switch to B/F/TAF. Are there any explanations for the decreases in albuminuria and beta-2-microglobulin and for the increases in spine BMD?

Minor comments

1. To be exact, trough concentration should be used, not trough
2. What do the author mean by lifestyle diseases?

Reviewer #2 (Comments for the Author):

Bicitegravir is an integrase strand transfer inhibitor (INSTI) used to treat HIV among people living with HIV (PLHIV). Bicitegravir properties related to potency and safety have been reported previously among PLHIV. However, the data on pharmacokinetics is limited among older PLHIV of whom physiologically due to age advancement, the existence of comorbidities, and chronic inflammatory status the pharmacokinetics may differ.

In the present article, the investigators assessed the pharmacokinetics of bicitegravir among 10 older male PLHIV in Japan. They report that all the pharmacokinetic parameters assessed were similar to those observed in younger HIV-negative Japanese. Indicating that the pharmacokinetics of bicitegravir is not affected by advancement in age among PLHIV.

The study has some strengths, the investigators have provided data for six important PK parameters for bicitegravir. However, there are some minor study design methodological issues to be addressed, it is unclear why only male patients were selected. Is the evidence in other ethnic populations suggests there is no difference between male and female bicitegravir pharmacokinetics?

The authors compared the PK of the study population with the young Japanese HIV-negative cohort and HIV-positive non-Japanese population in other countries, but the authors did not discuss much in comparing with HIV positive young Japanese population. It would have been better to have a comparison with this subpopulation.

In addition, AUC Cmax, and Tmax in this study are shown to be slightly higher, and T1/2 slightly longer compared to findings from the Caucasian population. However, the authors have not discussed this discrepancy and its clinical implication.

Staff Comments:

Preparing Revision Guidelines

To submit your modified manuscript, log onto the eJP submission site at <https://spectrum.msubmit.net/cgi-bin/main.plex>. Go to Author Tasks and click the appropriate manuscript title to begin the revision process. The information that you entered when you first submitted the paper will be displayed. Please update the information as necessary. Here are a few examples of required

updates that authors must address:

Please return the manuscript within 60 days; if you cannot complete the modification within this time period, please contact me. If you do not wish to modify the manuscript and prefer to submit it to another journal, please notify me of your decision immediately so that the manuscript may be formally withdrawn from consideration by Microbiology Spectrum.

Bictegravir is an integrase strand transfer inhibitor (INSTI) used to treat HIV among people living with HIV (PLHIV). Bictegravir properties related to potency and safety have been reported previously among PLHIV. However, the data on pharmacokinetics is limited among older PLHIV of whom physiologically due to age advancement, the existence of comorbidities, and chronic inflammatory status the pharmacokinetics may differ.

In the present article, the investigators assessed the pharmacokinetics of bictegravir among 10 older male PLHIV in Japan. They report that all the pharmacokinetic parameters assessed were similar to those observed in younger HIV-negative Japanese. Indicating that the pharmacokinetics of bictegravir is not affected by advancement in age among PLHIV.

The study has some strengths, the investigators have provided data for six important PK parameters for bictegravir. However, there are some minor study design methodological issues to be addressed, it is unclear why only male patients were selected. Is the evidence in other ethnic populations suggests there is no difference between male and female bictegravir pharmacokinetics?

The authors compared the PK of the study population with the young Japanese HIV-negative cohort and HIV-positive non-Japanese population in other countries, but the authors did not discuss much in comparing with HIV positive young Japanese population. It would have been better to have a comparison with this subpopulation.

In addition, AUC C_{max}, and T_{max} in this study are shown to be slightly higher, and T_{1/2} slightly longer compared to findings from the Caucasian population. However, the authors have not discussed this discrepancy and its clinical implication.

Point-by-point responses to Reviewers' comments:

Reviewer #1 (Comments for the Author):

The authors investigated the pharmacokinetics of bicitgravir among 10 Japanese people living with HIV-1 (PLWH) who were men aged 50 years or older 4 weeks after stable switch to coformulated bicitgravir, emtricitabine, and tenofovir alafenamide. The study is interesting; however, the sample size is too small. I have a few comments for the authors to consider.

Response: Thank you for your positive review of our manuscript, which we have revised according to the reviewer's recommendations as specified below.

Major comments:

1. While the study was designed as a pharmacokinetic investigation, the sample size of 10 patients is too small. As shown in Figure 2, the variability of trough concentration of bicitgravir is wide among the enrolled participants aged between 50 years and 80 years.

Response: Thank you for raising this important point. The observed variability in trough and peak concentrations is likely attributable to one patient (Patient 10) who showed a low BIC concentration at all time points (Figure 2 and newly added supplementary data 1). We have provided the PK parameters of each participant as supplementary data and relevant information in the Results section (lines 114–116).

2. Did the author investigate the intracellular concentration of tenofovir diphosphate among the participants?

Response: Assessment of the intracellular concentration of tenofovir was unfortunately out of scope for the current analysis, but highly relevant for future studies.

3. The conclusion on safety and efficacy among PLWH aged 50 years or older could not be safely reached from this single-arm study consisting of only 10 PLWH. Similarly, I am afraid that the conclusion that PK parameters were not correlated with the age of the participants enrolled can only be made with a larger sample size.

Response: Thank you for this valuable comment. We have highlighted this study limitation in the Discussion section of the revised manuscript (lines 176–177).

4. In the cited dolutegravir study (ref 8) regarding the PK parameters and adverse effects, PLWH who were aged 60 years or older were enrolled; questionnaire interview was

conducted to assess the neuropsychiatric aspects and sleep quality. In this current study, no such assessment was conducted.

Response: Our study did not include a questionnaire interview to assess neuropsychiatric aspects and sleep quality. This has been clarified in the revised manuscript, along with the addition of details on neuropsychiatric symptoms and sleep quality from the medical records (Results section, lines 128–131).

5. The authors are encouraged to provide more information on dietary supplements of the participants, not just those prescribed medicines for chronic diseases. Many dietary supplements taken by the participants might contain divalent or trivalent cation and might not be known to the treating physicians unless specifically inquired.

Response: It was confirmed at enrollment that participants were not taking any interacting drugs or supplements. The revised manuscript has been updated with details of the enrolment process, and the enrollment confirmation form has been included as supplemental data (lines 220–225).

6. The tubular functions in this study were assessed by albumin:creatinine ratio. The author are encouraged to provide data on urinary beta-2-microglobulin:creatinine ratio, which will be a better parameter for assessment of tubular function than urinary albumin:creatinine ratio.

Response: Data on the urinary β 2-microglobulin to creatinine ratio have been added to the revised manuscript (Figure 4B), with details included in the Results section (lines 143–146).

7. There were six participants who were receiving TAF-containing regimens before switch to B/F/TAF. Are there any explanations for the decreases in albuminuria and beta-2-microglobulin and for the increases in spine BMD?

Response: We are unable to suggest an appropriate explanation for the decrease in albuminuria. Information on the ART regimen before switching to BIC/F/TAF has been added to the revised manuscript (lines 146–152). Of note, spine BMD was not significantly changed after switching BIC/F/TAF in our study population.

Minor comments

1. To be exact, trough concentration should be used, not trough

Response: This has been updated in the revised manuscript.

2. What do the author mean by lifestyle diseases?

Response: Clarification on the lifestyle diseases observed in our study population has been added to the revised manuscript (Results section, lines 95–97).

Reviewer #2 (Comments for the Author):

Bictegravir is an integrase strand transfer inhibitor (INSTI) used to treat HIV among people living with HIV (PLHIV). Bictegravir properties related to potency and safety have been reported previously among PLHIV. However, the data on pharmacokinetics is limited among older PLHIV of whom physiologically due to age advancement, the existence of comorbidities, and chronic inflammatory status the pharmacokinetics may differ.

Response: Thank you for your positive review of our manuscript. We acknowledge that there are limited data on bictegravir pharmacokinetics among older PLHIV, and that pharmacokinetics may differ as a result of comorbidities, chronic inflammatory disease, and other confounding factors. We have added supplementary figures on the correlations of body weight or eGFR with PK parameters to the revised manuscript (supplementary data 2), with a comment in the Results section (lines 118–119) and a note on potential confounding factors in the Discussion (lines 169–173).

In the present article, the investigators assessed the pharmacokinetics of bictegravir among 10 older male PLHIV in Japan. They report that all the pharmacokinetic parameters assessed were similar to those observed in younger HIV-negative Japanese. Indicating that the pharmacokinetics of bictegravir is not affected by advancement in age among PLHIV. The study has some strengths, the investigators have provided data for six important PK parameters for bictegravir. However, there are some minor study design methodological issues to be addressed, it is unclear why only male patients were selected. Is the evidence in other ethnic populations suggests there is no difference between male and female bictegravir pharmacokinetics?

Response: The proportion of females among HIV-infected individuals is low in Japan. For example, of the 624 newly diagnosed HIV cases in Japan in 2021, only 10 were female (of which 3 were aged 50 or older). It is therefore extremely challenging to collect sufficient data on female individuals with HIV. However, previous studies show that the PK parameters of BIC are not affected by sex (as described in the package insert). This study limitations has been added (with references) to the Discussion section (lines 195–200).

The authors compared the PK of the study population with the young Japanese HIV-negative cohort and HIV-positive non-Japanese population in other countries, but the authors did not discuss much in comparing with HIV positive young Japanese population. It would have been better to have a comparison with this subpopulation.

Response: BIC PK data from a younger HIV-positive population are not available from previous reports; therefore, we used PK data from a younger HIV-negative population for comparison in our study. We have added this as a study limitation in the Discussion section (lines 195–200).

In addition, AUC Cmax, and Tmax in this study are shown to be slightly higher, and T1/2 slightly longer compared to findings from the Caucasian population. However, the authors have not discussed this discrepancy and its clinical implication.

Response: We added details on the relationship between BIC PK and ethnicity, body weight, and physique to the Discussion section of the revised manuscript (lines 169–174).

January 31, 2023

Dr. Koji Watanabe
Kokuritsu Kenkyu Kaihatsu Hojin Kokuritsu Kokusai Iryo Kenkyu Center
AIDS Clinical Center
1-21-1 Toyama
Shijuku-ku
Tokyo 162-8655
Japan

Re: Spectrum05079-22R1 (Pharmacokinetics of bictegavir in older Japanese people living with HIV-1)

Dear Dr. Koji Watanabe:

Your manuscript has been accepted, and I am forwarding it to the ASM Journals Department for publication. You will be notified when your proofs are ready to be viewed.

Sincerely,

Takamasa Ueno
Editor, Microbiology Spectrum
